# The development of Vi-Connect: An educational game for the social inclusion at school of students with vision impairment

Ifigeneia Manitsa[1]*, Maria Livanou[2], Stephanie Burnett Heyes[1], Fiona Barlow-Brown[3], Nuria Gardia[1], Olivia Siegfried[1], Zoe Clarke[1], Holly Coelho[1], Alberto De Caro[4]

1 Institute for Mental Health, School of Psychology, University of Birmingham, Birmingham, United Kingdom, 2 Institute of Psychiatry, Psychology & Neuroscience, King's College London, London, United Kingdom, 3 Department of Psychology, Kingston University London, London, United Kingdom, 4 Creative Technology Department, Faculty of Science and Technology, Bournemouth University, Bournemouth, United Kingdom

☙ These authors contributed equally to this work.
* i.manitsa@bham.ac.uk

**Data Availability Statement:** There are ethical restrictions which prevent public sharing of minimal data for this study. The full dataset

## Abstract

Students with vision impairment experience multiple social emotional challenges at school which stem from communication difficulties in the social relationships they develop with teachers and classmates. This study took a multi-method, multi-informant participatory approach to develop "Vi-Connect: A Social School Journey", a digital intervention in the form of an educational game aimed at promoting school social inclusion of students with vision impairment by scaffolding advocacy and social communication skills. The study consisted of three phases: Co-production before developing Vi-Connect (Phase 1), development of the prototype (Phase 2) and assessment of Vi-Connect (Phase 3). Four and five adolescents with vision impairment participated in Phases 1 and 3 of the study respectively. A second group of young people from Eye-YPAG (N = 8), a youth advisory group for eye and vision research, participated in Phase 1. Six professionals in the field of vision impairment participated in Phases 1 and 3. Reflexive Thematic Analysis was used to analyse the findings of Phases 1 and 3. A prototype of Vi-Connect including three school-based scenarios was developed based on the Curriculum Framework for Children and Young People with Vision Impairment and the experiences and suggestions of all participants. The findings of this study indicated that digital interventions can be an additional supportive educational tool to support the social inclusion of students with vision impairment, as they can facilitate coping with challenging social situations and promote self-advocacy. This is one of the first studies to involve students with vision impairment in intervention development, prioritising the lived experiences of this student population. Additionally, acknowledging the key role of professionals in the school inclusion of students with Special Educational Needs and Disabilities, this study involved professionals in the field of vision impairment. This research opens the field for the development of more accessible educational interventions for students with sensory impairments.

includes transcribed, potentially identifying conversations with participants. Data are available upon request from the University of Birmingham Ethics Committee via email (ethics-queries@contacts.bham.ac.uk) for researchers who meet the criteria for access to confidential data.

**Funding:** This research has been supported by the Public Engagement Fund from the University of Birmingham. The funders had no role in study design, data collection and analysis, decision to publish, or preparation of the manuscript.

**Competing interests:** The authors have declared that no competing interests exist.

## Introduction

Sensory impairment frequently affects an array of domains such as social interactions, social skills and behavioural patterns placing this population at risk for poor wellbeing and for developing mental health conditions such as depression and anxiety [1, 2]. Adolescents with vision impairment experience significant barriers in school in relation to social acceptance, social adaptation, academic performance, peer relationships and self-esteem, mutually affecting their sense of belonging [3]. They also struggle to fit in the wider school community often resulting in smaller peer networks or isolation [4]. Additionally, considering the vulnerabilities inherent in the period of adolescent devleopment, such as identity formation [5] coupled with cognitive, emotional, and biological changes [6], this appears to be a particularly vulnerable population that may require specialist support to overcome social and emotional challenges.

Recent research suggests that COVID-19 has also had a significant negative effect on the mental health [7] and social interactions of people with vision impairment [8], especially those with severe sight impairment (previously "blindness" according to the UK classification system). A study by Heinze et al. [9] focusing on the long-term effects of COVID-19 on adults with vision impairment showed increased levels of loneliness compared to typically developing populations. The limited research into the impact of COVID-19 on the mental health of children and adolescents with Special Educational Needs and Disabilities (SEND) has associated lockdowns and distance learning with an increased number of psychological threats (e.g., social isolation and emotional neglect) and reduced coping mechanisms in this group [10]. Previous research has also highlighted the negative effects of COVID-19 on the general anxiety levels and adaptive behaviours of children and adolescents with SEND with many participants displaying maladaptive behaviours [11]. Thus, it is expected that the above findings will also apply to the student population with vision impairment who were also susceptible to the significant effect of COVID-19 on educational design and delivery. Considering the above findings have highlighted the detrimental effect of COVID-19 on the social emotional development of individuals with vision impairment, there is an urgent need to address these challenges and provide appropriate educational support tailored to the specific needs of people with vision impairment.

Although this is the first study aiming to develop a serious game targeting school inclusion in young people with vision impairment, previous literature underscores the positive role of digital interventions in the form of educational games on social competence [12], self-esteem development and communication [13], and emotional regulation [14]. Digital games have been utilised as primary and secondary interventions across natural settings such as schools and home showing promising outcomes in the social and cognitive abilities of students with neurodevelopmental and SEND such as dyslexia, Autism Spectrum Disorder (ASD), and ADHD [15]. A review of serious games for autistic children has underscored the positive effect of such interventions on the expression of emotions and social engagement [16]. More research on the design and development of serious games for autistic children has shown their positive impact on the development of social skills and interactions with typically developing peers [17]. Further, a research study focusing on the design of a serious game aimed at improving the independent living skills of people with intellectual disability and ASD highlighted the benefits of co-produced learning interventions in the forms of serious games for people with SEND [18]. Although the focus of this prior study was not on vision impairment, the design framework of the aforementioned intervention which is based on the elements of participatory research, self-learning and personalisation can be particularly useful for future studies aimed at designing and developing serious games for students with SEND. The findings of a meta-analytic review showed that serious games have the potential to advance knowledge, reduce

mental health difficulties and find applications with diverse groups [19]. Another systematic review suggested that serious games can enhance social skills in classroom settings supporting students to practice and master social skills in low-risk environments [20].

The serious game platform developed in the current study resulted from a joint endeavour among academic researchers and professionals specialising in child and adolescent wellbeing, game developers, two Advisory Groups of adolescents with vision impairment, and one adult Advisory Group of educational professionals and experts in the field of vision impairment. Previous research highlights the central role of multidisciplinary collaboration in the development and advancement of serious games [21]. Adolescents with sensory impairments (vision impairment, hearing impairment and multi-sensory impairment or deafblindness) have rarely been included in intervention design and development. Co-production methodologies provide a unique opportunity for underrepresented populations to actively engage with and participate in research as equal collaborators by sharing their views and lived experiences [22]. A recent study by Giannakopoulos et al. (2018) aimed at developing a series of serious and inclusive games for children and young people with vision impairment aged 7–10 years showed the positive effects of participatory research and co-production with people with lived experience and professionals in the development and evaluation of such interventions [23].

To our knowledge, currently there are no interventions for adolescents with vision impairment focusing on supporting social communication, social relationships, and self-advocacy skills in the mainstream school environment. School is a social environment in which positive interaction with teachers and other staff members is critical to learning. The development of positive social relationships with typically developing peers is also particularly important as previous research has shown that a lack of school belonging undermines academic achievement [24, 25]. Developing positive social relationships is even more crucial in mainstream secondary school settings where students interact with many teachers, so such negotiations must be repeated many times throughout the day.

## The current study

This study aimed to initiate the development of an accessible educational game to address the social emotional, communication and self-advocacy needs of adolescents aged 12–16 years with vision impairment prioritising social relationships with teachers and classmates in school, in line with the recommendations of Area 9 "Health: Social, Emotional, Mental and Physical Wellbeing" of the Curriculum Framework for Children and Young People with Vision Impairment (CFVI). The socio-ecological model that has been developed by the first author of the study [3, 26] based on Bronfenbrenner's ecological systems theory [27] has informed elements of Vi-Connect, which emphasises school belonging as well as peer and teacher support as significant social factors in the social emotional development of adolescents with vision impairment [3]. The proposed socio-ecological model provides a conceptual framework to understand and promote the social inclusion of students with vision impairment in school with a specific focus on the social (social relationships and participation in school activities), cultural (country of residence and education system) and chronological (age of the individuals) factors that may affect school belonging. Our previous research also highlighted the importance positive social relationships with peers and staff in the school belonging of adolescents with vision impairment [3]. Therefore, the main purpose of Vi-Connect is to promote the school-based social relationships of students with vision impairment through the development of social skills that can have a positive impact on such relationships.

A range of social skills were targeted in this study based on the CFVI and previous literature on social emotional challenges experienced by adolescents with vision impairment in school

(see [24, 25, 28–32]. Based on the CFVI and previously reviewed literature focusing on the social challenges faced by students with vision impairment, the specific social skills that Vi-Connect targeted are the following: self-advocacy, problem solving skills, contact initiation, making and maintaining relationships, self-efficacy and agency, and confidence to interact with others independently. Three scenarios were developed that aimed to improve these social skills by promoting social relationships with teachers and peers. Similar to the design framework of other studies focusing on the design and development of educational games for students with SEND [18, 23], the main purpose of the study was to initiate and promote self-learning, personalisation and continuous challenge.

Finally, people with vision impairment in the UK are categorised into two groups according to their registration status: those with severe sight impairment (previously "blindness" according to the UK classification system) and those with sight impairment (previously "low vision" according to the UK classification system). Considering the additional adaptations that students with severe sight impairment may require, the focus of this study was on students with sight impairment only. Further, students with intellectual disabilities and complex needs did not take part in this study due to the potential impact of those additional needs on their ability to participate in an online focus group and in the implementation of the educational intervention. Considering the diverse needs and different types of social interactions that occur in specialist educational settings, students from special schools did not participate in this study. Given current evidence for educational interventions targeting the social emotional support provided to students with vision impairments is mainly drawn from research on younger children [4] and considering the challenges specific to adolescence discussed above, the focus of the current study was to develop an intervention with and for adolescents with vision impairment currently attending mainstream schools only.

## Method

### Participants

Three advisory groups participated in the present study: a professional group and two youth groups. All participants were based in the UK at the time of the study. All adults participating in the study were required to have professional experience working with adolescents with vision impairment. No specific years of experience with adolescents with vision impairment were required for the study. Some professionals also had lived experience of vision impairment (that of a close relative). Participants for the professional advisory group were recruited through social media, twitter, and an online forum for professionals in the field of vision impairment. Seven female professionals took part overall ($Mage$ = 51.86; $SD$ = 12.23) with six taking part in Phase 1 ($Mage$ = 53.67; $SD$ = 12.32) and six taking part in Phase 3 ($Mage$ = 50.00; $SD$ = 12.26). The experience of the professionals participating in the research spanned a broad range of relevant areas: Qualified Teachers of the Vision Impaired (QTVIs), Ophthalmology, National Organisations and Charities, Educational Psychologists.

All adolescents in the study had to be registered as sight impaired (previously "with low vision" according to the UK classification system), aged 12–16 years and attending mainstream schools. Adolescents with additional needs and from special schools did not participate in this study. Participants for the youth advisory groups fell into two categories: an online group and an in-person group. In Phase 1, the online group consisted of four adolescents (Female 1, 20%, Male 3, 60%, aged 12–16 years; $Mage$ = 14.25, $SD$ = 1.71). In Phase 3, the online group of adolescents with vision impairment included five participants (Female 2, 40%; Male 3, 60%; aged 12–16 years; $Mage$ = 14.4, $SD$ = 1.52). All participants in both online groups were registered as partially sighted or with low vision. The in-person group that participated in Phase 1 consisted

of eight members from Eye-YPAG, a young person's advisory group of 8–16-year-olds with a range of vision problems, including refractive errors.

Pseudonyms have been used for all participants.

The study received ethical approval from the University of Birmingham's Science, Technology, Engineering and Mathematics Ethics Review Committee on 18th May 2023 (approval number: ERN_0898-May2023). The recruitment period for this study was between 18 May (immediately after ethical approval was obtained) and 27 June 2023, when the last advisory group meeting took place.

## Design and procedure

The current study used a model of a multi-method, multi-informant participatory approach, to create a prototype digital intervention in the form of an educational game for young people with vision impairment. The study featured three Phases: (1) designing the digital intervention, (2) developing the content of the digital intervention, and (3) assessing the digital intervention. All participants gave both written and verbal consent to participate in the study, and parents of adolescents also gave written consent for their child's participation.

Focus groups were conducted with each advisory group in Phases 1 and 3 of the study. In Phase 1, the first advisory group meeting was held in-person with Eye-YPAG members (aged 8–16 years). The first author of the study led a 45-minute focus group which was facilitated by one of the research assistants of the project *(insert initials)*. Subsequently, a 90-minute online focus group was held with an advisory group of professionals in the field of vision impairment, led by the first author of the study *(insert initials)*, and facilitated by one of the co-authors *(insert initials)*. Finally, a 90-minute focus group with an advisory group of young people with vision impairment was held online with four participants. This focus group was also led by the first author of the study and facilitated by one of the co-authors *(insert initials)*. Participants in all online groups were paid £25 for their time and sessions were recorded via Microsoft Teams Platform with the permission of participants to enable transcription. A fee was also paid for collaboration with Eye-YPAG.

In Phase 2, a prototype 2D Educational game "Vi-Connect: A Social School Journey" was developed by the game developer *(insert initials)* from the research team. Initially, the first author and the research assistants of the research project *(insert initials)* had drafted potential scenarios for the game based on previous literature on the school inclusion of adolescents with vision impairment and the social challenges they experience in the school environment as well as the proposals of Area 9 of CFVI. Following Phase 1 and as a result of young people's input and feedback, these initial draft scenarios were partially changed. The most significant change was that, following advice from the advisory groups, more focus was centred on peer relationships within each of the scenarios.

Once the educational game had been developed, an assessment phase took place (Phase 3). Online focus groups were conducted with both the professional advisory group and with the online group of young people with vision impairment. Participants were sent an email with a link to the game, and they were informed about the tasks they would complete in the second meeting. Both assessment sessions with the advisory groups took 90 minutes and participants were paid £25 for their time.

Please see Fig 1 for steps undertaken to co-design and develop Vi-Connect.

## Materials

Four different interview schedules were developed for the purposes of this study: two for the focus groups with adolescents (Phases 1 and 3) and two for the focus groups with professionals

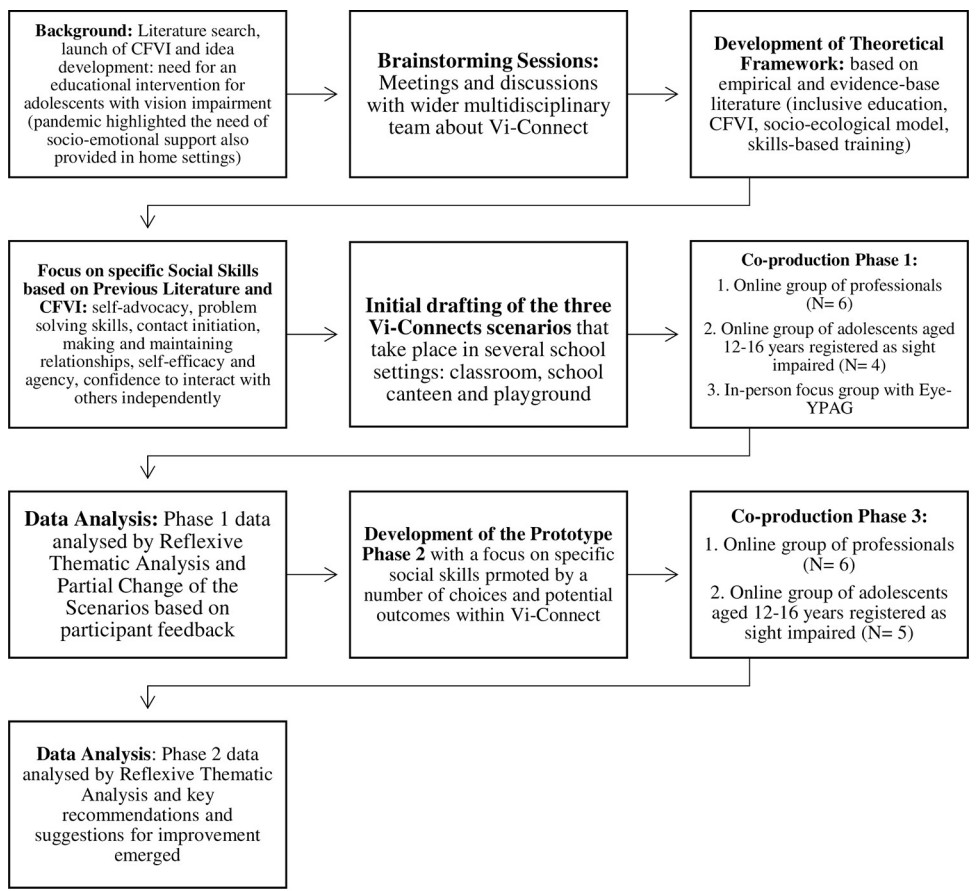

**Fig 1. Research, co-production, design and assessment of Vi-Connect.**

(Phases 1 and 3). For the Phase 1 focus groups with adolescents, questions and prompts were given about the school day (classroom and breaktime experiences and relationships with teachers and peers), video game preferences (types of games and platform preferences), accessibility features that are useful and suggestions for the content of an educational game focused on school inclusion. At the end of the focus group, Eye-YPAG members completed a short survey with four open-ended questions about the content of the game, the platforms they use to play games and potential accessibility features. For the Phase 1 focus group with professionals, questions and prompts were given around storyline suggestions, characters, game rewards and accessibility features. Before the meeting with professionals, a list of potential accessibility features was circulated for discussion in the meeting.

For the assessment phase, questions were framed around the content of the game (thoughts about the story, characters, graphics, text, length, and pace of the game); the impact of the educational game intervention (positive and negative feelings about each scenario); whether participants felt that the game would help in school life; accessibility of the game and any suggestions for improving the game intervention.

## Data analysis

Reflexive Thematic Analysis emphasises a flexible process of coding data and identifying themes building upon Braun & Clarke's original six steps of thematic analysis [33]. The first step (1) involves familiarisation with the data by repeatedly reading the transcripts, followed

by (2) initial coding of key features of the data, (3) searching for and generating possible themes and sub themes, (4) checking the identified themes against the data, (5) naming and defining the identified themes and finally (6) writing up the results [34].

The recordings of the advisory group meetings were transcribed by the four research assistants of the team *(initials)*. Four research assistants then analysed the transcriptions independently, dividing the transcripts between them, under the close supervision of the first author (steps 1 and 2). Potential themes and sub-themes were discussed during regular team meetings between the four research assistants and the first author (step 3). Additional meetings between the first four authors were held to review the identified themes and sub-themes and complete the analysis (steps 4 and 5). The four research assistants provided drafts of the results section with the most representative quotes under the supervision of the first author. The first four authors refined the analysis and added it to the manuscript (step 6).

## Results

### Phase 1: Co-production before developing Vi-Connect

Participants were positive about co-designing a digital intervention that would enhance the social emotional skills of adolescents with vision impairment and provided useful suggestions towards its development. Four main themes were developed and have been further analysed below: "Social Experiences Within School", "Video Game Preferences", "Content of the Intervention" and "Accessibility Features".

**Social experiences within school.**   The school experiences of adolescents with vision impairment revealed a profound desire for full participation in all academic activities and a deep aspiration for their future studies. However, their learning and inclusion in the school environment are frequently disrupted by communication challenges with school staff and the pervasive lack of proper adaptations. Several adolescents conveyed that they often struggle with articulating their needs. This difficulty is compounded by any co-existing mental health challenges and a pervasive lack of comprehension from both teachers and peers.

"*I think I do struggle a lot to ask for help in classrooms because people like to stare at you. . . Everyone asks like, why has to have a sheet bigger? And sometimes you just don't want to explain because it's quite upsetting when people ask you all the time.*" (James, 12 years, male)

Professionals also recounted the challenges that students with vision impairment face in school settings. These were generally characterised by social exclusion, with some professionals noting that students with vision impairment rarely engage with their peers during break times and prefer to socialise with their teaching assistant on group tasks. Others mentioned that some school procedures intended to support students with vision impairment can also result in further exclusion from academic activities.

"*(. . .) because they don't want to be different or they, you know, there's so many other layers to how they feel about the visual impairment that we're not very good at dealing with (. . .)*" (Samantha, 53 years, female, QTVI)

The pervasive lack of understanding within schools deeply affects the relationships adolescents with vision impairment develop with both teachers and peers. The quality of relationships with teachers is often contingent upon the extent of assistance they provide and the degree of understanding they display. This not only impacts the interpersonal dynamics but also influences other areas of their school life, such as their academic performance.

*"Some teachers do understand, and I've had work sent to me through a Chromebook instead so I can enlarge the documents to how, how I can read it, which has been good, but at the same time, sometimes they, sometimes they don't, and that can, that can, make things a bit difficult in lessons. . ."* (Mike, 15 years, male)

However, it's noteworthy that amidst these challenges, there are several glimmers of hope and support. Some adolescents with vision impairment find solace in friendships that genuinely understand and cater to their unique needs.

*"I've got a good group of friends, so if they see me, they'll come to me first, which is really nice of them."* (Alice, 14 years, female)

**Video game preferences.** All participants were positive about developing an educational game for the social inclusion of students with vision impairment and emphasised the need to focus on difficult social situations that take place at school. Adolescents expressed a strong affinity for playing videogames and were enthusiastic about the prospect of co-developing an educational game. Participants typically engaged with games on popular platforms but also highlighted the appeal of an accessible online game, emphasising its potential compatibility with larger PC screens and zoomable tablets.

*"You could play it on a computer and tablet because they're bigger screens and they can be touch screens".* (Mark, 15 years, male)

The most popular games were based on real life events and situations, where players could undertake many engaging adventures and compelling narratives. Adolescents highly valued games as a means of learning, with participants sharing that videogames could help them improve coordination, everyday life skills, and enhance their classroom learning.

*"Some games might not be educational, but they can help you with coordination and skills. They can make you better at sports or improve multi-tasking".* (Laura, 15 years, female)

Professionals' attitudes towards educational games were also positive and their group listed several advantages of implementing such games in a school setting. They noted that educational games can provide a safe place for students with vision impairment to practice challenging situations and could also be used as a tool to improve understanding among sighted students.

**Content of the intervention.** Adolescents provided insightful feedback on various elements of the game, encompassing aspects such as the title, Main Character, and scenarios included. Regarding the title, adolescents highlighted the necessity for the game's title to strike a harmonious balance between being descriptive yet engaging.

*"I think the name should be alliterative so it can stick, and people will have a positive reading and tell their friends about it"* (Alan, 16 years, male)

When asked about the key characteristics of the game, adolescents and professionals noted that it should include challenging scenarios that students with vision impairment encounter in real life. Adolescents provided several suggestions for potential scenarios, such as how to build social relationships within school, how to manage social situations such as bullying, or how to enhance their social skills by learning how to ask peers or teachers for help.

*"I really struggle to ask the teachers because I have a lot of mental health issues. . . So sometimes it's really hard for me to ask for help which is something that I really struggle with, especially like when the class is silent and asking in front of a lot of people it's, it's erm, awkward and very difficult for me"* (James, 12 years, male)

Participants agreed that the game should be engaging and encourage students with vision impairment to practice their skills regularly. Notably, many professionals believed that the game should be accessible to teachers and sighted peers alongside students with vision impairment. They argued that this would promote students' perspective-taking abilities and place less responsibility on the student with vision impairment to change their behaviour:

*"(. . .) helping other people around them to understand their perspective more means that they're more likely to encourage them to do these things in that situation and it takes the emphasis of them having to, you know, always be the one that makes that change."* (Ella, 46 years, female, educational psychologist)

Both adolescents and professionals agreed that the Main Character should be customisable in terms of physical appearance suggesting that this would make the game feel more personal to players. However, some professionals raised concerns over the extent to which customisation options could accurately reflect all players and suggested non-specific characters instead. Although most adolescents and professionals believed that the Main Character should be visually impaired, several professionals endorsed the idea of giving players the option to play as a sighted character.

*"It would be quite powerful, wouldn't it? If say they got to choose whether they're visually impaired or not."* (Jane, 53 years, female, QTVI)

When asked about specific scenarios to be included in the game, many professionals mentioned that unstructured periods at school are often challenging for students with vision impairment. They explained that the level of support received from teaching assistants during classes was often reduced at lunch or break times. Professionals also noted that including teamwork scenarios would be valuable, arguing that students with vision impairment commonly feel excluded from group discussions. Some professionals emphasised the importance of including challenging situations within structured teaching periods, such as asking a teacher for additional support.

*"Finding the language to explain it to your peers in the playground or on the sports field is just as important as reminding a teacher or explaining to a supply teacher that you need your work adapted and how you need it adapted and why you need it adapted."* (Deanne, 63 years, female, executive manager of a UK-based charity)

**Accessibility features.** Participants mentioned a number of accessibility features that may be particularly useful to players with vision impairment, such as high-quality audio, clearly differentiable voices, text descriptions, and reducing background noise during interactions with other characters. Professionals recommended focusing on the other senses of students with vision impairment to enhance their gaming experiences. One professional remarked that haptic feedback during pertinent parts of the game could be beneficial for some of the students she supports:

*"I've got a few young people on my caseload who game and some of them have no usable vision and rely heavily on the vibrate function (. . .)"* (Amy, 36 years, female, QTVI trainee)

Further exploration revealed that adolescents appreciate a high degree of personalisation in accessibility features. Their specific suggestions surrounding font types, sizes, and colour contrast underscored the idea that accessibility is highly individualistic, emphasising the importance of allowing players to tailor these features to their unique requirements. Audio descriptions and text-to-speech functionalities, particularly delivered via interactive speech bubbles, were also notably favoured by the group.

*"The text in the game should be very easily readable. Also, there should be a colour change option for colour blind participants."* (Laura, 15 years, female)

### Phase 2: Development of the prototype

After conducting the three focus groups, the game developer of this study started working on the development of the digital intervention. "Vi-Connect: A Social School Journey" (see Fig 2) is a 2D Educational game that is accessible through Windows devices in which players with vision impairment can create their own gender-neutral Main Character (see Fig 3) and participate in three different scenarios that take place in various school settings. Our research team tried to develop as representative scenarios as possible related to the social emotional experiences of students with vision impairment, therefore the three stories we further developed and used in the development of the current prototype take place in three different school settings: classroom, school canteen and playground. During these scenarios, the Main Character interacts with their classmates and school staff, such as teachers, teaching assistants and canteen staff.

On first entering the game, the player builds their own personalised Main Character. Players are asked to choose the external characteristics of their Main Character, such as hair colour

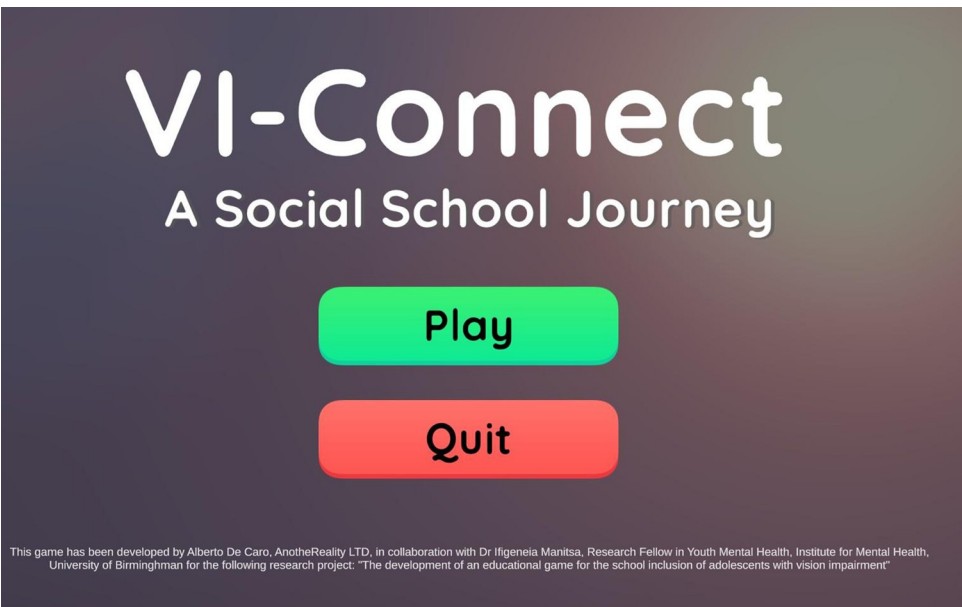

**Fig 2. The start screen of Vi-Connect: A social school journey.**

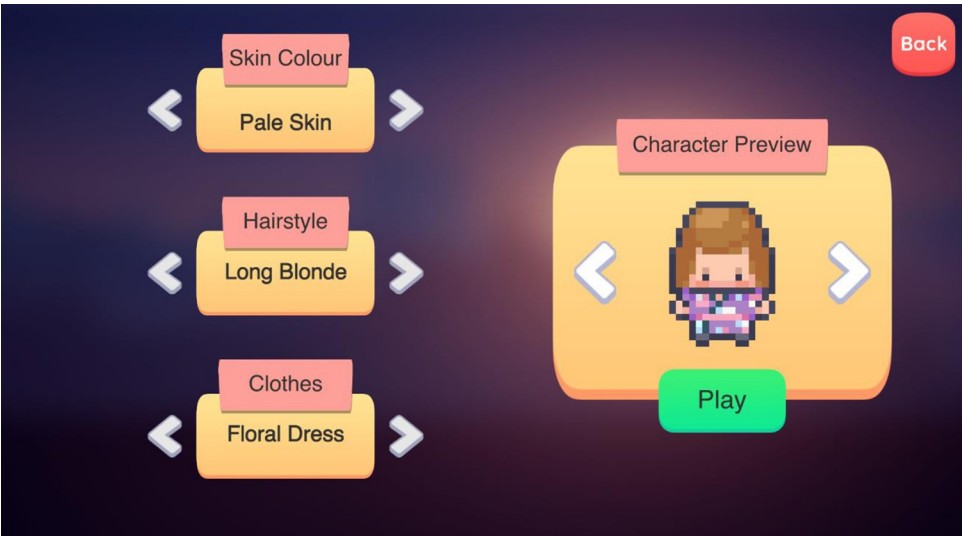

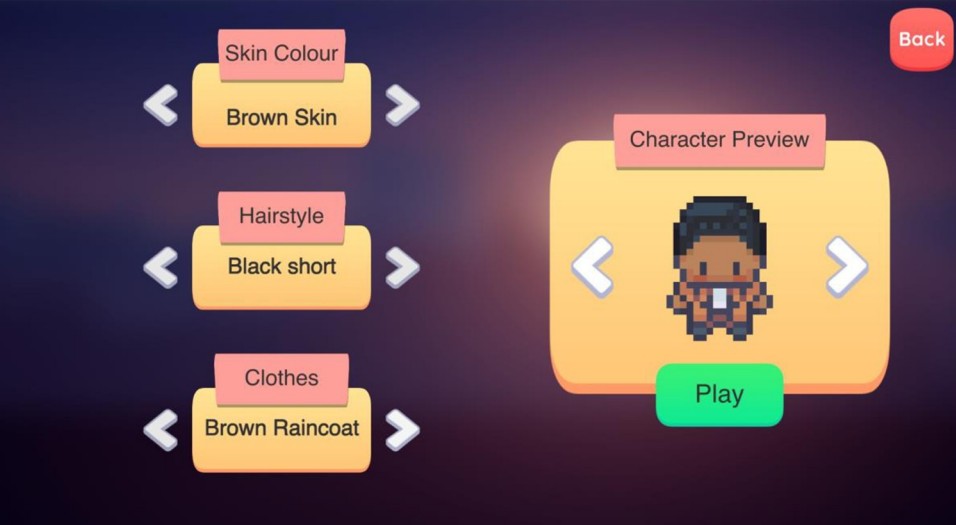

**Fig 3. Examples of the character customisation screen.** Players can create their own Main Character and choose their skin colour, hairstyle, and clothes.

(clearly distinguishable colours with clear contrast) and clothes. Then, the Games Master presents a set of instructions to players describing the process of the game.

Six choices are given to the player after the presentation of each scenario (see Figs 4 and 5 for specific examples), who must decide how the Main Character will react to a certain situation that takes place in a school setting. After the player makes their choice, in the Outcome phase the Games Master congratulates the players on their choice and provides them with additional feedback. Players are not criticised for the choices they made, as there are no "right", or "wrong" choices included in this game. Players can make their own choices that match their personality, as well as past and current school experiences. The Games Master's feedback is constructive and aims to support students with similar dilemmas in real-world situations. Although the Games Master's feedback is not negative or invasive, they attempt to

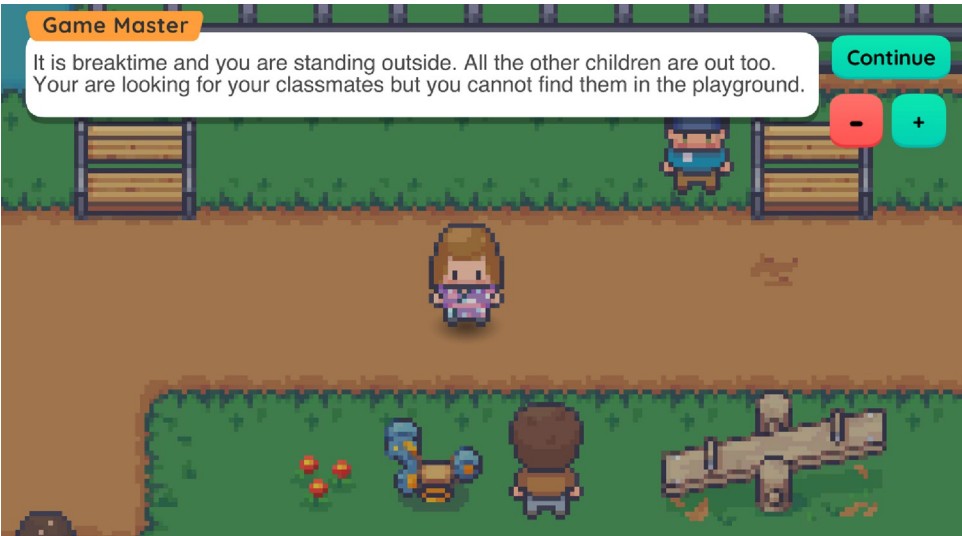

**Fig 4. Examples of the first screen of the third game scenario.** This scenario focuses on the social experiences of students with vision impairment in their school playground.

promote through their advice the development of social relationships in the school environment and increase the confidence of the players in initiating social interactions with different people within their school environment. At the end of scenarios one and two, the Games Master prepares the player for the next scenario.

Building on previous literature that has highlighted the social challenges that students with vision impairment experience at school and in line with the CFVI, Vi-Connect aims to promote the social skills development of young people with vision impairment and to help them cope with a range of social-emotional challenges that appear in their school environment with an emphasis on the social relationships they develop with their classmates and school staff. The purpose of each scenario is to help young people with vision impairment increase their self-advocacy skills, feel confident discussing their vision impairment with people outside their

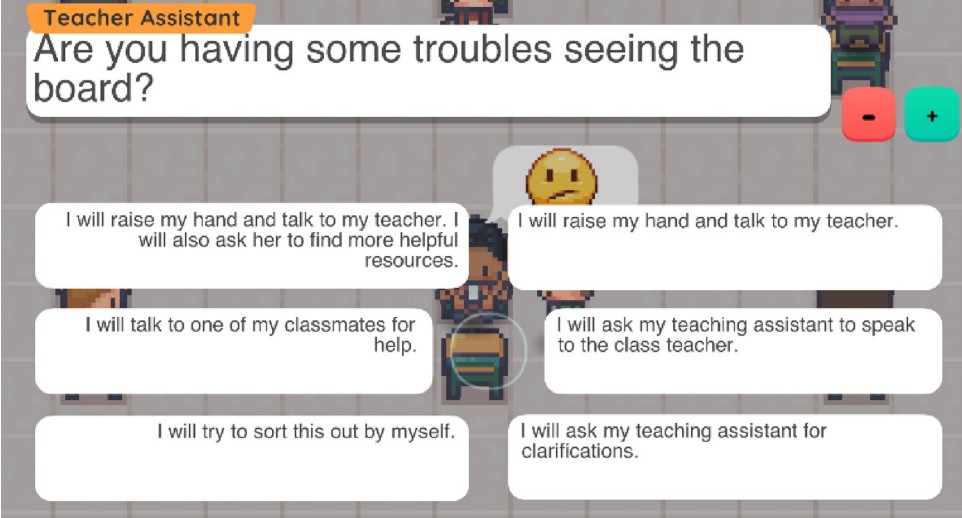

**Fig 5. The six choices for the first game scenario.** This game scenario focused on the social experiences of students with vision impairment in the classroom. Six choices are given to the player after the presentation of each story.

immediate circle (e.g., school staff and classmates) and identify an appropriate person to ask for support when needed. Specifically, as discussed in the Introduction, Vi-Connect targets the following social skills: self-advocacy, problem solving skills, contact initiation, making and maintaining relationships, self-efficacy and agency, and confidence to interact with others independently. A list of the scenarios and available choices as well as potential outcomes (Games Master feedback) are presented in Table 1.

## Levels and aesthetics

The game is 2D top-down educational game that uses pixel art asset packs from the Unity Asset Store.

Considering feedback from adolescents and professionals, the levels use high contrast background (black background whenever possible) and adjustable font size (see Fig 6) to ease the distinction of different elements on the screen.

**Mechanics and UI.** The main mechanics of the game are Dialogue Interaction between Player and Non-Player Characters (NPCs) and Movement. The Interaction is automatically triggered by the game, and this leads to the opening and closing of the dialogue boxes.

Following suggestions from Phase 1 participants and to promote player understanding of the game, each line of dialogue plays an audio transcript when the text appears. Based on the participants' suggestions, the user can increase or decrease the font size of the text displayed in the dialog boxes. To move forward to the next line, the player must select the Continue button using their mouse/trackpad. A similar approach is used for the Choices, where each choice is a button that can be selected.

All the UI elements follow the SCULPT Accessibility guidelines (SCULPT for Accessibility | Worcestershire County Council).

Movement is handled by pressing the W/A/S/D buttons on the keyboard.

## Phase 3: Assessment of Vi-Connect

Participants (adolescents with vision impairment and professionals) shared positive feedback about the game, including its concept, purpose, and content. They offered suggestions for improvement to fit the differing needs of students with vision impairment. Three themes were identified and presented below: "Storyline of Vi-Connect", "Accessibility Features" and "Impact of the Intervention".

**Storyline of Vi-Connect.** Adolescents and professionals commented on the degree of personalisation, relatability of the scenarios, realistic choices, and potential for developing the game further. They explained that the game involved a lot of customisations, that they enjoyed being able to personalise the Main Character and adjust the font size, and that there was a good variety of choices to choose from in the scenarios. Adolescents felt that the game was relatable and represented their challenges with vision impairment in a realistic manner, whilst also delivering effective and beneficial advice.

> "*I think the turnout of the game is really good and I think it's got a good message. The sounds are really good the way you can increase the font size, really good, the accessibility for the game is very good. It's quite clear as well.*" (James, 12 years, male)

Mixed opinions emerged regarding the number of choices in the game. Some participants believed that the number of choices was high and could confuse players. However, others felt that the number of choices was sufficient and provided comprehensive coverage of the range of social situations that young people with vision impairment struggle with.

**Table 1. Vi-Connect scenarios, choices and potential outcomes.**

| School Setting | Description | Task | Choices | Games Master Feedback |
|---|---|---|---|---|
| **Classroom** | The English teacher writes some letters on the board. The Main Character cannot see the letters. | Identify the best person to help them. | **Choice 1:** Ask their teaching assistant to explain the letters on the screen. | **Outcome 1:** The player should speak to their teaching assistant about the additional support they require. |
| | | | **Choice 2:** Ask their teaching assistant if they can talk to their class teacher about this issue to help them find additional resources that can help them complete and future classroom tasks. | **Outcome 2:** The player should speak to their class teacher about the additional support they require. |
| | | | **Choice 3:** Explain to the class teacher that they cannot see the screen. | **Outcome 3:** The player should explain their needs to their class teacher. |
| | | | **Choice 4:** The class teacher suggests discussing this later, but the Main Character insists to do it now as they cannot follow the lesson. | **Outcome 4:** The Games Master congratulates the player for choosing to talk to their class teacher. |
| | | | **Choice 5:** Ask a classmate sitting nearby for additional support. | **Outcome 5:** The player should talk to their class teacher. |
| | | | **Choice 6:** Prefer not to talk with anyone. | **Outcome 6:** The player should explain their needs to school staff and classmates. |
| **School Canteen** | The Main Character attempts to find their classmates. | Decide who is the right person to ask for support. | **Choice 1:** Prefer not to talk to anyone. One of their peers approaches them. | **Outcome 1:** Self-advocacy can be rewarding as they can receive the support they need. |
| | | | **Choice 2:** Ask their friend to help them find a table. Their friend initiates contact with their classmates. | **Outcome 2:** The player should initiate contact, and talk to the other children on the table, even if these are not from their class. |
| | | | **Choice 3:** Ask their friend to help them find a table. The Main Character initiates contact with their classmates. | **Outcome 3:** The player should consider sitting next to and talking to their classmates. |
| | | | **Choice 4:** Locate a table with the assistance of a canteen staff member. The canteen staff accompanies the Main Character. The Main Character initiates contact with their classmates. | **Outcome 4:** The player should talk to more people and sit at different tables as this will help them make more new friends. |
| | | | **Choice 5:** Locate a table with the assistance of a canteen staff member. The Main Character approaches the table without the canteen staff's support. The Main Character initiates contact with their classmates. | **Outcome 5:** Similar to Outcome 4, the player should talk to more people and sit at different tables as this will help them make more new friends. |
| | | | **Choice 6:** Locate a table with the assistance of a canteen staff member. The canteen staff accompanies the Main Character. The canteen staff initiates contact with their peer group. | **Outcome 6:** The player should initiate contact and talk to the other children on the table. |
| **School Playground** | The Main Character cannot find their classmates in the playground. | Decide who is the right person to ask for support in the playground. | **Choice 1:** Ask a peer to play with them. | **Outcome 1:** The player can approach a bigger group of friends next time. |
| | | | **Choice 2:** Ask a peer to help them find their classmates. The Main Character would like the peer to initiate contact with their classmates. | **Outcome 2:** It might be good for the Main Character to approach their classmates next time, as this will help them make new friends. |
| | | | **Choice 3:** Ask a peer to help them locate their classmates. The Main Character waits for their classmates to initiate contact with them. | **Outcome 3:** The player should speak to their classmates and initiate contact with them. |
| | | | **Choice 4:** Locate their classmates with the help of a peer. Their classmates do not respond to their initial question. | **Outcome 4:** The Main Character's classmates are busy playing, therefore they might have missed their initial question. |
| | | | **Choice 5:** Attempt to locate their classmates on their own. One of the peers approaches them and helps them locate their peer group. | **Outcome 5:** Self-advocacy can be very rewarding as they can receive the support they need. |
| | | | **Choice 6:** Ask a staff member to help them approach their classmates. The Main Character is happy to talk to their peers as well as making initial contact. | **Outcome 6:** The player should also approach other classmates in the future. |

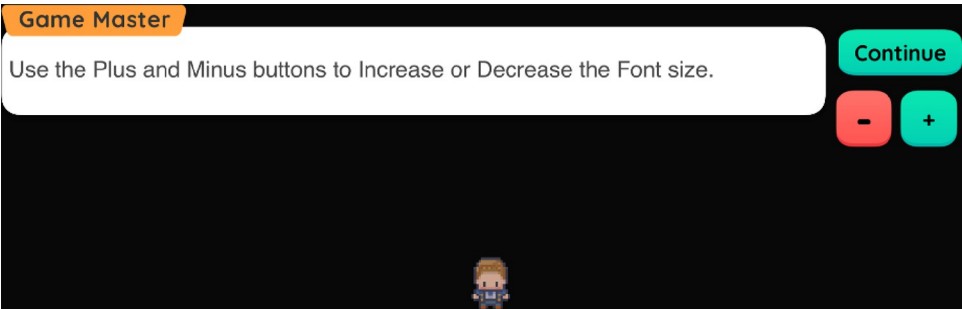

**Fig 6. Adjustable font size.** Players have the option to adjust the font size of the game. The Games Master gives them all the appropriate instructions.

*"I think that like three or four would be good to make sure you can keep your attention on it for."* (Lily, 14 years, female)

Adolescents had various ideas for how the content could be improved in the game, regarding the scenarios. They suggested that there could be an independent revision session scenario for students with vision impairment who approach GCSEs or A-levels. This would then make it useful and applicable to older students and help those who may be struggling in these specific situations. Other suggestions regarding the content of the game mainly involved adding more personalisation, such as being able to choose the Main Character's name and having less computerised voices.

**Accessibility features.** Participants explained that they were able to access the game and follow along easily. The ability to change and alter the font size of the text was said to be a good and useful feature, making the game accessible regardless of their diverse visual needs.

*"I like (that) I can make the font size bigger, like, on your own,"* (Alice, 14 years, female)

There were also some suggestions the adolescents gave regarding how to make it more accessible based on their own experiences and by envisioning how other students with vision impairment may struggle. One example given was adding audio description of visual elements of the game for those with severe vision impairment.

*"I like it when they are moving like he walks over to his friends at a bench or something. And obviously I myself wouldn't need that. But if it's someone that is blind or something like that, accessibility is such a big thing on apps and stuff nowadays that you need to cover everything."* (James, 12 years, male)

Games Master was considered a very useful feature of the game as it read aloud the instructions and choices for each scenario, allowing players with vision impairment to better navigate the game and be less stressed.

Both adolescents and professionals suggested that players should be able to choose their preferrable background colours, as what suits one may hinder another's ability to play.

*"...but I just thought something slightly bolder and an option for having a different colour background like... some children work with prefer like a yellow background on the right end, just to make it easier."* (Natalie, 41 years, female, QTVI)

**Impact of the intervention.** Both adolescents and professionals thought that this game could be a very helpful tool to use in schools. This is because it can promote the social skills of students with vision impairment, by showing them how to better manage particularly stressful or difficult situations that could arise because of their vision impairment.

"*Probably the classroom one as it is most relatable. . . if I'm sat at the front of the room, I still can't see the board and there was only one teacher who 'understood'*" (Alan, 16 years, male)

According to the participants, one of the main strengths of the game is that it allows players to replay each scenario and make a different choice each time they play. Through this process, players with vision impairment can become familiar with possible response options they may not be comfortable to try in their daily life but can practice "safely" in the game.

Professionals reported that the current intervention would be helpful to promote the self-advocacy skills of students with vision impairment and the social relationships they develop with their sighted peers. They explained that Vi-Connect can help children and young people with vision impairments to better articulate their needs to their immediate environment, such as school staff and close friends. They also mentioned that such educational interventions can be particularly useful for young people's transition to Higher Education and employment, as they can further promote their independence skills.

"*We have to think that the points we're trying to get about self-advocacy and being able to explain yourself and interact. . . It about what they're having to cope with and what they're having to explain in some way how they want to be more accepted and how eventually they want to be able to explain themselves formally and informally to their workplace, teachers, to their mates.*" (Rose, 71 years, female, ex-optometrist)

Some adolescents and professionals suggested the game should be made more applicable to younger children to increase its impact (Key Stages 1 and 2), with more difficult levels or modules included for older adolescents. Professionals explained that it would be useful to develop a series of different social stories within the school settings that could be applied both to children and adolescents with vision impairment depending on their age and other clinical factors, such as the presence of additional needs.

"*And just whether this would appeal to the older children towards the age of 16 or maybe you have when you go into the first page, the option of going with the younger version or the older version? Once you've selected that, then take that then takes you to age-appropriate scenarios.*" (Natalie, 41 years, female, QTVI)

## Discussion

To our knowledge this is the first study that initiated the development of an educational game on the social emotional needs and inclusion of students with vision impairment in secondary mainstream settings, addressing a gap in previous literature on games to promote the social emotional development of young people with sensory impairments (vision impairment, hearing impairment and multi-sensory impairment or deafblindness). Although previous literature has highlighted the positive effects of educational interventions on the development of social skills [31], there appears to be a distinct lack of educational interventions aimed at promoting

the communication and self-advocacy skills of adolescents with sensory impairments within school [35–37].

Vi-Connect builds on the CFVI principles and areas suggested Douglas et al. (2019) [37] of fostering autonomy and agency in this population with digital technology by targeting the learning areas proposed: independence skills, social emotional functioning, and school inclusion. Our proposed digital intervention is also in line with the basic principles of inclusive education and social learning theory. In particular, the key principle of inclusive education is that all students should be included in the same classrooms and learn with and from each other despite their different needs and abilities [38]. Considering the key proposals of inclusive education, the main goal of the suggested intervention is to improve the social interactions and relationships that adolescents with vision impairment develop with their teachers and classmates in mainstream schools and promote their social inclusion. In addition, through observing of the Main Character of Vi-Connect and the impact their choices can have on their school life, players with vision impairment are expected to learn how to make beneficial choices that promote their wellbeing and school inclusion. The self-learning component of Vi-Connect derives its key features from Bandura's social learning theory, which proposes that observation and imitation promote the development of prosocial behaviours [39].

Our study followed a multi-method, multi-informant participatory design, as students with vision impairment often report a lack of participation in decision-making about their educational plans and school inclusion [24]. Also acknowledging the considerable role that teacher support and encouragement play in the lives of adolescents with vision impairment [25, 28, 29], the current research aimed to take a holistic approach, actively involving professionals. All the suggestions made by the participants in Phase 1 were incorporated into the current version of Vi-Connect (Phase 2) after having first been discussed with the multidisciplinary research team and the game developer. Specifically, participants requested that Vi-Connect include real-life situations that could relate to the school life of students with vision impairment and promote the development of their social skills. As suggested by our participants, the Main Character is customisable. Adolescents also chose the title of the current intervention and decided that the term "vision impairment" should be included in it. Regarding the accessibility features included in Vi-Connect, according to participant feedback, high quality audio, clear voices, simple graphics (e.g., black background) and reduced background noise have been included in the game.

The key findings of Phases 1 and 3 are that students with vision impairment enjoy using games as supplementary educational tools that can positively affect their academic and social experiences at school. Such interventions can further promote social emotional learning in school and support school staff who often indicate a perceived lack of confidence in implementing educational interventions for the emotional well-being of students with vision impairment [40]. However, it is important to recognise that such interventions can only be used as supplementary tools to support the school inclusion of vulnerable student populations, as inclusive education practice must be directly linked to the school curriculum and continuous teacher training. In particular, teacher training interventions have been linked to more positive teacher attitudes and increased knowledge about inclusion and continuing skill development [41].

All the participants also commented very positively on the accessibility features of Vi-Connect and its specific focus on the individual needs of students with vision impairment, a finding which is in line on previous research on the development of educational games as supplementary tools to promote youth mental health [14]. Such findings may be particularly useful to other researchers and developers of educational games for students with vision impairment, as they may offer insights into the lived experiences of these students and the

social elements that should be prioritised in such games. The adolescents who participated in this study reflected on their school experiences and reported the social emotional challenges they face at school. Such challenges could potentially be the main focus of future digital interventions aimed at promoting the school inclusion of children and young people with vision impairment. Our participants also referred to key features of the game that promoted the accessibility of Vi-Connect (e.g., audio, graphics, font size). Such accessibility features can be particularly useful to other game developers who design educational games for people with vision impairment. The co-production approach we took to developing Vi-Connect highlights the importance of involving people with lived experience in developing such games.

This is one of the first studies that involved students with vision impairment in intervention development and planning. Despite the transformational effects of co-production and participatory research to service development [42] evident in the current study that prioritised the lived experiences of adolescents and professionals in the field of vision impairment, there are limitations that should be further discussed and suggestions for future research. First, Vi-Connect in its current form consists of three stories that focus on the social relationships that students with vision impairment develop at school, therefore other relevant scenarios and challenges that arise in the school environment are not included in this prototype. Second, some participant suggestions and feedback from Phase 3 were not embedded into the current version of Vi-Connect. For example, adolescents mentioned that their teachers should also play the game as this would help them better understand their needs. Previous research has shown that implementing disability awareness interventions among non-disabled individuals can increase tolerance and acceptance of people with disabilities and diverse needs [43]. This suggestion could potentially be incorporated into a multiplayer version of Vi-Connect where students with vision impairment would be invited to play alongside their sighted peers. Multiplayer educational games have been found to have more positive effects on learning outcomes compared to single-player games [44]. Third, as previously discussed, the focus of the current game was adolescents with sight impairment meaning that the players had some residual vision. Thus, more adaptations are required for student populations with more severe types of sight impairment. Also considering that adolescents with intellectual disabilities and complex needs were not involved in the development and assessment of Vi-Connect, the current prototype has not been designed for special schools. From our findings, it is evident that future research should prioritise adapting and customising Vi-Connect to meet the diverse needs of students with vision impairment and to be applicable in different educational contexts. Fourth, the current study focused on a very specific student population with vision impairment, including five students aged 12–16 who were registered as sight impaired (or "with low vision"). Furthermore, only a limited number of professionals participated in the study and all our participants were based in the UK at the time of the interview. Considering, therefore, the small size of the study and cultural differences that might have significantly influenced our findings [31], it would be useful to conduct cross-cultural studies with larger samples sizes and more diverse populations in terms of age and special needs. It would also be useful to conduct longitudinal studies to assess the effectiveness of Vi-Connect over time and its long-term impact on the social inclusion of students with vision impairment.

Our aim is to evaluate the effectiveness of Vi-Connect and trial it with different student groups with and without additional sensory impairments (e.g., hearing impairment and multisensory impairment or deafblindness) in different geographical locations and schools. To trial Vi-Connect with different populations and assess its long-term impact, our research team is considering a range of quantitative and qualitative research methodologies that can be tailored to the learning needs of each student population, such as interviews and focus groups, large surveys and observational research. We are interested in conducting longitudinal studies with

diverse groups of students to examine the effect of such digital interventions on their social competence, school engagement and academic performance. The proposed research methods will also help us understand whether students attending more well-resourced schools are more likely to have positive mental health and social outcomes compared to young people from deprived areas. Applying a range of different research methods may further promote mixed methods approaches to the development of educational interventions for students with SEND as well as open the research field to designing more effective interventions for students with sensory impairments who have been under-represented in previous research.

Overall, the main purpose on this study was to develop a fully accessible educational game for students with vision impairment taking into full account their own voices and needs. This study attempted to address the existing gap in previous literature and to initiate the implementation of the recommendations included in the CFVI focusing on the social emotional development and school inclusion of students with vision impairment. The majority of previous research explores the social and emotional challenges that students with vision impairment face in school without offering solutions to address them. As previously discussed, most intervention studies focus on younger children only, therefore current interventions focusing on adolescents with vision impairment are very limited [31] and none of them are digital interventions. The majority of current educational interventions also take place in the school environment [37] which can cause many problems in their implementation due to significant budget and time constraints. Parents of students with other SEND such as ASD often report their children's limited access to mental health support [45]. It has also been reported that only 17% of individuals experiencing vision loss report receiving some type of social emotional support in response to their needs [46]. Thus, our goal was to develop a prototype of an inclusive educational intervention that is also fully accessible in home settings. In order for teachers and other school staff (e.g., teaching assistants and special educators) to promote Vi-Connect in the classroom, they do not need specialised training but access to a Windows device and an Internet connection. Students who wish to engage with Vi-Connect also do not need to receive special training or have previous experience playing video games, as there are detailed instructions and support (e.g. Games Master figure) to guide and help them to complete the intervention and also boost their confidence. The proposed intervention aims to promote inclusivity by being both user and family friendly. Our research increased knowledge about the unique school experiences of students with vision impairment, a student population significantly underrepresented in previous research and practice. Previous research has also highlighted the positive outcomes of co-designing psychosocial interventions with education providers for the school inclusion of students with SEND [47], a key feature of the current study. Thus, it is expected that the proposed digital intervention and its key features will open the field for the development of more accessible educational interventions for children and young people with sensory impairments that will further promote their agency and independence [48] through the active involvement of young people and professionals in their development and evaluation.

## Acknowledgments

We are grateful for the ideas and discussions with adolescents and professionals. We are also grateful for the ideas and discussions with the Eye-YPAG. The Eye-YPAG is funded by Moorfields Eye Charity, the NIHR BRC at Moorfields eye hospital and Institute of Ophthalmology, and Santen. We would also like to thank Ms Sophie Temple for proofreading the paper.

## Author Contributions

**Conceptualization:** Ifigeneia Manitsa, Maria Livanou.

**Data curation:** Ifigeneia Manitsa, Maria Livanou, Stephanie Burnett Heyes, Nuria Gardia, Olivia Siegfried.

**Formal analysis:** Ifigeneia Manitsa, Maria Livanou, Stephanie Burnett Heyes, Nuria Gardia, Olivia Siegfried, Zoe Clarke, Holly Coelho.

**Funding acquisition:** Ifigeneia Manitsa, Maria Livanou, Stephanie Burnett Heyes.

**Investigation:** Ifigeneia Manitsa, Maria Livanou, Stephanie Burnett Heyes, Fiona Barlow-Brown, Nuria Gardia, Olivia Siegfried.

**Methodology:** Ifigeneia Manitsa, Maria Livanou, Stephanie Burnett Heyes, Zoe Clarke, Holly Coelho.

**Project administration:** Ifigeneia Manitsa, Maria Livanou, Stephanie Burnett Heyes, Fiona Barlow-Brown, Nuria Gardia, Alberto De Caro.

**Resources:** Ifigeneia Manitsa, Alberto De Caro.

**Software:** Ifigeneia Manitsa, Alberto De Caro.

**Supervision:** Ifigeneia Manitsa, Maria Livanou, Stephanie Burnett Heyes, Fiona Barlow-Brown.

**Validation:** Ifigeneia Manitsa.

**Visualization:** Ifigeneia Manitsa.

**Writing – original draft:** Ifigeneia Manitsa, Maria Livanou, Stephanie Burnett Heyes, Fiona Barlow-Brown, Nuria Gardia, Olivia Siegfried, Zoe Clarke, Holly Coelho, Alberto De Caro.

**Writing – review & editing:** Ifigeneia Manitsa, Maria Livanou, Stephanie Burnett Heyes, Fiona Barlow-Brown.

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
