## [Decision Letter · Decision Letter 0]

11 Apr 2024

PONE-D-24-06103The Development of Vi-Connect: An Educational Game for the Social Inclusion at School of Students With Vision ImpairmentPLOS ONE

Dear Dr. Manitsa,

Thank you for submitting your manuscript to PLOS ONE. After careful consideration, we feel that it has merit but does not fully meet PLOS ONE’s publication criteria as it currently stands. Therefore, we invite you to submit a revised version of the manuscript that addresses the points raised during the review process.

We look forward to receiving your revised manuscript.

Kind regards,

Tien-Chi Huang

Academic Editor

PLOS ONE

Journal Requirements:

   "This research has been supported by the Public Engagement Fund from the University of Birmingham."

4. In this instance it seems there may be acceptable restrictions in place that prevent the public sharing of your minimal data. However, in line with our goal of ensuring long-term data availability to all interested researchers, PLOS’ Data Policy states that authors cannot be the sole named individuals responsible for ensuring data access (http://journals.plos.org/plosone/s/data-availability#loc-acceptable-data-sharing-methods).

Additional Editor Comments:

I am pleased to inform you that after thorough evaluation by two expert reviewers, your manuscript titled "The Development of Vi-Connect: An Educational Game for the Social Inclusion at School of Students With Vision Impairment" (Manuscript ID: PONE-D-24-06103) has been considered for publication in PLOS ONE. The decision is Minor Revision.

The reviewers commend your work's research value and relevance but recommend specific enhancements to strengthen the manuscript.

Your attention to these suggestions will significantly enhance your manuscript's impact and relevance. We look forward to your revised submission and are hopeful for a positive outcome.

Reviewers' comments:

Reviewer's Responses to Questions

**Comments to the Author**

1. Is the manuscript technically sound, and do the data support the conclusions?

Reviewer #1: Yes

Reviewer #2: Yes

2. Has the statistical analysis been performed appropriately and rigorously? 

Reviewer #1: Yes

Reviewer #2: Yes

3. Have the authors made all data underlying the findings in their manuscript fully available?

Reviewer #1: No

Reviewer #2: Yes

4. Is the manuscript presented in an intelligible fashion and written in standard English?

Reviewer #1: Yes

Reviewer #2: Yes

5. Review Comments to the Author

Reviewer #1: Review Report

Dear authors, thank you for submitting your manuscript to our journal. While it is a commendable piece of work, we offer a series of comments to enhance the quality of your manuscript.

Manuscript: "The Development of Vi-Connect: An Educational Game for the Social Inclusion at School of Students With Vision Impairment"

Theoretical Framework:

Suggested Improvement: Expand the theoretical review to include comparative studies of similar tools in populations with other disabilities, to contextualize the uniqueness and added value of Vi-Connect.

Update of Sources:

Suggested Improvement: Include more recent studies on the impacts of the pandemic on the education of students with visual impairments, strengthening the justification of the study in the current context.

Appropriateness of Objectives:

Suggested Improvement: Refine the objectives to include specific goals related to the socio-emotional skills that Vi-Connect aims to develop, providing clarity in the game's expectations.

Methodology:

Suggested Improvement: Detail the participant selection process and the inclusion/exclusion criteria to strengthen the methodological validity.

Ethical Aspects:

Suggested Improvement: Describe the measures taken to ensure the accessibility and usability of the game for participants with visual impairments, ensuring equity in participation.

Discussion:

Suggested Improvement: Deepen the discussion on how Vi-Connect aligns with the principles of inclusive education and social learning theory.

Suggested Improvement: Analyze the implications of the findings for developers of digital educational tools, offering recommendations based on the study's experience.

Conclusions:

Suggested Improvement: Clarify how Vi-Connect addresses the deficiencies identified in the theoretical framework, directly linking the study's results with the stated objectives.

Suggested Improvement: Include reflections on the scalability and sustainability of Vi-Connect in different educational settings.

Limitations:

Suggested Improvement: Explicitly acknowledge the limitations related to the study design, such as sample size and geographical scope, and their impact on the generalization of the results.

Future Prospects:

Suggested Improvement: Propose longitudinal studies to evaluate the effectiveness of Vi-Connect over time and its long-term impact on the social inclusion of students.

Suggested Improvement: Suggest future research focused on adapting and customizing the game to meet diverse needs and educational contexts.

General Conclusion:

The manuscript presents an innovative and relevant study with the potential to positively impact the inclusion of students with visual impairments. To strengthen the study's contribution, we recommend an expansion and deepening in certain methodological, theoretical, and analytical aspects, as well as a more detailed discussion on the practical implications of the findings and future research directions.

Reviewer #2: This is a valuable and worthwhile piece of research that has truly involved young people with vision impairment. It is good to note that there has been the genuine involvement of Eye-YPAG as well as other participants as part of the development of a digital intervention to support the social inclusion of students with vision impairment.

The authors have provided a strong rationale for this research that is clear and valid.

The methods are clearly outlined and the ethical considerations are thorough. The procedures are technically sound and the research enables the team to achieve their aims. This research is very specific and the conclusions are supported by the data gathered.

Just a few points to consider:

Where Bronfenbrenner’s ecological systems theory is mentioned (line 95) it would be helpful to explain how this theory has informed elements of Vi-Connect. This is not clarified later in the description of the game.

The presentation of the themes and the selected quotes from the participants around their social experiences within school provide important data (aside from the development of a digital intervention tool).

The construction of the game is outlined clearly and the scenarios are well-described. The participants have been enabled to respond to the game and suggest improvements.

Page 44 (line 51) There is a full stop missing after (3):

Adolescents with vision impairment experience significant barriers in school in relation to social acceptance, social adaptation, academic performance, peer relationships and self-esteem, mutually affecting their sense of belonging (3).

Line 52: There is a full stop missing after (4):

They also struggle to fit in the wider school community often 52 resulting in smaller peer networks or isolation (4)

Line 61: There is a full stop missing after (10):

Serious games have been utilised as primary and secondary 59 interventions across natural settings such as schools and home showing promising outcomes in 60 neurodevelopmental and Special Educational Needs and Disabilities (SEND) such as dyslexia, 61 autism, and ADHD (10)

Line 90: Please would it be possible to reference this: Curriculum Framework for Children and Young People with Vision Impairment (CFVI) (i.e. Hewett, R., Douglas, G., McLinden, M., James, L., Brydon, G., Chattaway, T., Cobb, R., Keil, S., Raisanen, S., Sutherland, C., Taylor, J. (2022). Curriculum Framework for Children and Young People with Vision Impairment (CFVI): Defining specialist skills development and best practice support to promote equity, inclusion and personal agency. RNIB)

Line 194: Typo: Tour research assistants then analysed the transcriptions 195 independently (Tour should read as Four).

Line 425: GCSE’s – no apostrophe required.

Line 499: CFVI principles. Reference Hewett et al. (2022) as above.

Line 533: participatory research to service development (30)evident – space missing

Line 683: Inconsistent referencing. This should be Douglas, G., McLinden, M etc

6. PLOS authors have the option to publish the peer review history of their article (what does this mean?). If published, this will include your full peer review and any attached files.

Reviewer #1: **Yes: **David Pérez Jorge

Reviewer #2: No

---

## [Author Response · Author response to Decision Letter 0]

16 May 2024

Response to the Journal and Reviewers

Journal Requirements

Thank you for this very useful comment. We have now made sure that our manuscript meets PLOS ONE's style requirements, including those for file naming.

Thank you for this comment. No funding information is included in the manuscript. The following sentence included in the Acknowledgments section refers to Eye-YPAG and we have been advised to include it in this section as it is their standard practice when participating in peer-reviewed publications.

3. Thank you for stating the following financial disclosure: "This research has been supported by the Public Engagement Fund from the University of Birmingham."

Thank you for this useful suggestion. This information has been now included in our cover letter.

4. In this instance it seems there may be acceptable restrictions in place that prevent the public sharing of your minimal data. However, in line with our goal of ensuring long-term data availability to all interested researchers, PLOS’ Data Policy states that authors cannot be the sole named individuals responsible for ensuring data access (http://journals.plos.org/plosone/s/data-availability#loc-acceptable-data-sharing-methods).

Thank you for this comment. As is usual for qualitative data we are not ethically permitted to share the full dataset including transcribed conversations with participants. In the study information sheets provided to participants we stated that all the information they provided will be stored on a local password-protected server accessible only to members of the research team. As such we are unable to share the data widely.

As previously mentioned, it has been clarified in the study information sheets that only members of the research team will have access to the data. In line with the ethics guidelines of the University of Birmingham and the ethics approval of the study, recordings and transcripts of this study cannot be shared with the public. During our focus groups with participants, very sensitive topics regarding the mental health of a very vulnerable student population were discussed. Further, considering the very specific population that this study focused on, our data can be highly identifiable.

6. Please review your reference list to ensure that it is complete and correct. If you have cited papers that have been retracted, please include the rationale for doing so in the manuscript text or remove these references and replace them with relevant current references. Any changes to the reference list should be mentioned in the rebuttal letter that accompanies your revised manuscript. If you need to cite a retracted article, indicate the article’s retracted status in the References list and also include a citation and full reference for the retraction notice.

Thank you for the helpful comment and suggestion. We have now checked and updated our reference list.

Response to Reviewers’ Comments

We would like to thank very much the reviewers for their very helpful comments and suggestions. Please read our responses below.

Reviewer #1: Review Report

Dear authors, thank you for submitting your manuscript to our journal. While it is a commendable piece of work, we offer a series of comments to enhance the quality of your manuscript.

Manuscript: "The Development of Vi-Connect: An Educational Game for the Social Inclusion at School of Students With Vision Impairment"

Thank you very much for all your comments and suggestions. Please find our responses below.

Theoretical Framework:

Suggested Improvement: Expand the theoretical review to include comparative studies of similar tools in populations with other disabilities, to contextualize the uniqueness and added value of Vi-Connect.

Thank you for your comments. Based on your suggestions, more studies and additional information have been added to paragraphs 3 and 4 of the Introduction. In particular, the following paragraphs have been added to the Introduction:

Paragraph 3: Although this is the first study that aims to develop a serious game for the school inclusion of young people with vision impairment, previous literature underscores the positive role of digital interventions in the form of educational games on social competence (12), self-esteem development and communication (13), and emotional regulation (14). Digital games have been utilised as primary and secondary interventions across natural settings such as schools and home showing promising outcomes in the social and cognitive abilities of students with neurodevelopmental and SEND such as dyslexia, Autism Spectrum Disorder (ASD), and ADHD (15). A review of serious games for autistic children has underscored the positive effect of such interventions on the expression of emotions and social engagement (16). More research on the design and development of serious games for autistic children has shown their positive impact on the development of social skills and interactions with their typically developing peers (17). Further, a research study focusing on the design of a serious game aimed at improving the independent living skills of people with intellectual disability and ASD highlighted the benefits of co-produced learning interventions in the forms of serious games for people with SEND (18). Although the focus of this prior study was not on vision impairment, the design framework of the aforementioned intervention which is based on the elements of participatory research, self-learning and personalisation can be particularly useful for future studies aimed at designing and developing serious games for students with SEND. The findings of a meta-analytic review showed that serious games have the potential to advance knowledge, reduce mental health difficulties and find applications with diverse groups (19). Another systematic review suggested that serious games can enhance social skills in classroom settings supporting students to practice and master social skills in low-risk environments (20).

Paragraph 4: A recent study by Giannakopoulos et al. (2018) aimed at developing a series of serious and inclusive games for children and young people with vision impairment aged 7-10 years showed the positive effects of participatory research and co-production with people with lived experience and professionals in the development and evaluation of such interventions (23).

Update of Sources:

Suggested Improvement: Include more recent studies on the impacts of the pandemic on the education of students with visual impairments, strengthening the justification of the study in the current context.

Thank you so much for this very helpful comment. Based on your suggestions, we have added a paragraph focusing on the effect of COVID-19 on the social emotional development of people with vision impairment and on the mental health of children and adolescents with Special Educational Needs and Disabilities. As mentioned in this paragraph, there is an apparent lack of studies focusing on the impact of COVID-19 on the social emotional development and inclusion of visually impaired children and youth. However, it is expected that the aforementioned findings will also apply to the visually impaired student population who were also sensitive to the significant impact of COVID-19 on educational design and delivery.

Based on the reviewer’s suggestions, the following paragraph has been added to the Introduction:

Recent research suggests that COVID-19 also had a significant negative effect on the mental health (7) and social interactions of people with vision impairment (8), especially the population with severe sight impairment (previously “blindness” according to the UK classification system). A study by Heinze et al. (2021) focusing on the long-term effects of COVID-19 on adults with vision impairment showed increased levels of loneliness compared to the typically developing population (9). The limited research of the impact of COVID-19 on the mental health of children and adolescents with Special Educational Needs and Disabilities (SEND) has associated lockdowns and distance learning with an increased number of psychological threats (e.g., social isolation and emotional neglect) and reduced coping mechanisms in these students (10). Previous research has also highlighted the negative effects of COVID-19 on the general anxiety levels and adaptive behaviours of children and adolescents with SEND with many participants displaying maladaptive behaviours (11). Thus, it is expected that the above findings will also apply to the student population with vision impairment who were also susceptible to the significant effect of COVID-19 on educational design and delivery. Considering the above findings that have highlighted the detrimental effect of COVID-19 on the social emotional development of individuals with vision impairment, there is an urgent need to address these challenges and provide appropriate educational support tailored to the specific needs of people with vision impairment.

Appropriateness of Objectives:

Suggested Improvement: Refine the objectives to include specific goals related to the socio-emotional skills that Vi-Connect aims to develop, providing clarity in the game's expectations.

Thank you for this comment. Based on your comments and the comments of the second reviewer, additional information has been added to the first two paragraphs of the “The Current Study” subsection to provide clarity on the expectations of the game:

This study aimed to initiate the development of an accessible educational game to address the social emotional, communication and self-advocacy needs of adolescents aged 12-16 years with vision impairment prioritising social relationships with teachers and classmates in school, in line with the recommendations of Area 9 “Health: Social, Emotional, Mental and Physical Wellbeing” of the Curriculum Framework for Children and Young People with Vision Impairment (CFVI). The socio-ecological model that has been developed by the first author of the study (Manitsa, 2023; Manitsa et al., 2023) based on Bronfenbrenner’s ecological systems theory (26) has informed elements of Vi-Connect, which emphasises school belonging as well as peer and teacher support as significant social factors for the social emotional development of adolescents with vision impairment (3). The proposed socio-ecological model provides a conceptual framework to understand and promote the social inclusion of students with vision impairment in school with a specific focus on the social (social relationships and participation in school activities), cultural (country of residence and education system) and chronological (age of the individuals) factors that may affect school belonging. Our previous research also highlighted the importance positive social relationships with peers and staff in the school belonging of adolescents with vision impairment (27). Therefore, the main purpose of Vi-Connect is to promote the school-based social relationships of students with vision impairment through the development of social skills that can have a positive impact on such relationships. 

A range of social skills were targeted in this study based on the CFVI and previous literature on social emotional challenges experienced by adolescents with vision impairment in school (see (24,25,28–32). Based on the CFVI and previously reviewed literature focusing on the social challenges faced by students with vision impairment, the specific social skills that Vi-Connect targeted are the following: self-advocacy, problem solving skills, contact initiation, making and maintaining relationships, self-efficacy and agency, and confidence to interact with others independently. Three scenarios were developed that aimed to improve these social skills by promoting social relationships with teachers and peers. Similar to the design framework of other studies focusing on the design and development of educational games for students with SEND (18,23), the main purpose of the study was to initiate and promote self-learning, personalisation and continuous challenge.

Further, additional information has been added to the “Phase 2: Development of the Prototype” section to remind the reader of the exact social skills that Vi-Connect aims to develop:

Specifically, as discussed in the Introduction, Vi-Connect targets the following social skills: self-advocacy, problem solving skills, contact initiation, making and maintaining relationships, self-efficacy and agency, and confidence to interact with others independently.

Methodology:

Suggested Improvement: Detail the participant selection process and the inclusion/exclusion criteria to strengthen the methodological validity.

Thank you for this comment. Additional information has been added to the last paragraph of the Introduction section (“The Current Study” subsection) and to the first two paragraphs of the Participants subsection.

Introduction: Finally, people with vision impairment in the UK are categorised into two groups according to their registration status: those with severe sight impairment (previously “blindness” according to the UK classification system) and tho

---

## [Decision Letter · Decision Letter 1]

11 Jun 2024

PONE-D-24-06103R1The Development of Vi-Connect: An Educational Game for the Social Inclusion at School of Students With Vision ImpairmentPLOS ONE

Dear Dr. Manitsa,

Thank you for submitting your manuscript to PLOS ONE. After careful consideration, we feel that it has merit but does not fully meet PLOS ONE’s publication criteria as it currently stands. Therefore, we invite you to submit a revised version of the manuscript that addresses the points raised during the review process.

**ACADEMIC EDITOR: **Thank you for your resubmission of manuscript number PONE-D-24-06103R1, titled "The Development of Vi-Connect: An Educational Game for the Social Inclusion at School of Students With Vision Impairment." We appreciate the effort you have put into addressing the reviewers' comments in the previous rounds of review. The manuscript has improved significantly. However, minor revisions are still needed to fully meet the journal's standards.

We look forward to receiving your revised manuscript.

Kind regards,

Tien-Chi Huang

Academic Editor

PLOS ONE

Journal Requirements:

Reviewers' comments:

Reviewer's Responses to Questions

**Comments to the Author**

1. If the authors have adequately addressed your comments raised in a previous round of review and you feel that this manuscript is now acceptable for publication, you may indicate that here to bypass the “Comments to the Author” section, enter your conflict of interest statement in the “Confidential to Editor” section, and submit your "Accept" recommendation.

Reviewer #1: All comments have been addressed

Reviewer #2: All comments have been addressed

2. Is the manuscript technically sound, and do the data support the conclusions?

Reviewer #1: Partly

Reviewer #2: Yes

3. Has the statistical analysis been performed appropriately and rigorously? 

Reviewer #1: Yes

Reviewer #2: Yes

4. Have the authors made all data underlying the findings in their manuscript fully available?

Reviewer #1: No

Reviewer #2: Yes

5. Is the manuscript presented in an intelligible fashion and written in standard English?

Reviewer #1: Yes

Reviewer #2: Yes

6. Review Comments to the Author

Reviewer #1: Manuscript PONE-D-24-06103R1

Dear authors,

I have reviewed the revisions made by the authors in response to the reviewers' comments for the manuscript titled "The Development of Vi-Connect: An Educational Game for the Social Inclusion at School of Students With Vision Impairment." Below is a detailed evaluation of the adequacy of the changes made and additional recommendations.

Evaluation of the Revisions Made

Theoretical Framework and Literature Review:

The authors have expanded the theoretical review to include comparative studies of similar tools in populations with other disabilities and added a paragraph on the impacts of the COVID-19 pandemic on the education of students with visual impairments. These changes provide a broader and stronger context for the research and reinforce the study's justification in the current context.

Study Objectives:

The study objectives have been refined to include specific goals related to the socio-emotional skills that Vi-Connect aims to develop. This addition improves the clarity and precision of the game's expectations.

Methodology:

The process for selecting participants and the inclusion/exclusion criteria have been detailed, enhancing the transparency and methodological validity of the study. Additionally, measures taken to ensure the accessibility and usability of the game for participants with visual impairments have been described, ensuring equity in participation.

Game Development and Content:

The authors have added details about the game's accessibility features based on participant feedback, ensuring the game's design is suitable for users with limited vision. They have also described the main character's features and the challenging social situations included in the game, which enhances the game's relevance and potential impact.

Results and Discussion:

The discussion has been expanded to align with the principles of inclusive education and social learning theory. The authors have provided practical implications and recommendations for developers of digital educational tools, enriching the interpretation of the results and offering a useful guide for future developments in the field.

Conclusions and Future Research:

It has been clarified how Vi-Connect addresses the deficiencies identified in the theoretical framework, providing a clearer connection between the results and the study's objectives. The authors have explicitly acknowledged the limitations related to the study design, such as sample size and geographical scope, and have proposed longitudinal studies to evaluate Vi-Connect's effectiveness over time.

Additional Recommendations

Despite the significant improvements made, I suggest the following minor modifications for further clarity and coherence:

Terminology Clarification:

Ensure that the terminology used to describe individuals with visual impairments is consistent throughout the manuscript. For example, use terms like "visual impairment" or "limited vision" uniformly.

Game Implementation Details:

Include more details on how the game will be implemented in real educational settings. This could involve information about the necessary training for teachers and technical support to ensure effective integration.

Citations and References:

Review and ensure that all citations and references are accurate and complete. Some minor typographical errors and inconsistencies in references should be corrected to maintain the academic accuracy of the manuscript.

Impact Evaluation:

Include a brief discussion on how the long-term impact of the game will be measured beyond the initial evaluations. This will help contextualize the game's effectiveness and sustainability in different educational settings.

Conclusion

Overall, the authors have adequately addressed all the reviewers' comments and suggestions. The changes made have significantly improved the clarity, coherence, and methodological rigor of the manuscript. With the additional minor modifications recommended, I suggest that the revised manuscript be considered for publication in PLOS ONE.

Sincerely,

Reviewer #2: Thank you for this resubmission. I have now had the opportunity to read through the revised manuscript. I am happy that the suggestions that I have made have been addressed.

7. PLOS authors have the option to publish the peer review history of their article (what does this mean?). If published, this will include your full peer review and any attached files.

Reviewer #1: **Yes: **Pérez-Jorge, D

Reviewer #2: No

---

## [Author Response · Author response to Decision Letter 1]

21 Jun 2024

Response to the Journal and Reviewers

Journal Requirements

Thank you for the helpful comment and suggestion. We have now checked and updated our reference list.

Response to Reviewers’ Comments

We would like to thank very much the reviewers for their very helpful comments and suggestions. Please read our responses below.

Reviewer #1: Manuscript PONE-D-24-06103R1

Dear authors,

I have reviewed the revisions made by the authors in response to the reviewers' comments for the manuscript titled "The Development of Vi-Connect: An Educational Game for the Social Inclusion at School of Students With Vision Impairment." Below is a detailed evaluation of the adequacy of the changes made and additional recommendations.

Evaluation of the Revisions Made

Theoretical Framework and Literature Review:

The authors have expanded the theoretical review to include comparative studies of similar tools in populations with other disabilities and added a paragraph on the impacts of the COVID-19 pandemic on the education of students with visual impairments. These changes provide a broader and stronger context for the research and reinforce the study's justification in the current context.

Thank you very much for your positive feedback.

Study Objectives:

The study objectives have been refined to include specific goals related to the socio-emotional skills that Vi-Connect aims to develop. This addition improves the clarity and precision of the game's expectations.

Thank you very much for your positive feedback.

Methodology:

The process for selecting participants and the inclusion/exclusion criteria have been detailed, enhancing the transparency and methodological validity of the study. Additionally, measures taken to ensure the accessibility and usability of the game for participants with visual impairments have been described, ensuring equity in participation.

Thank you very much for your positive feedback.

Game Development and Content:

The authors have added details about the game's accessibility features based on participant feedback, ensuring the game's design is suitable for users with limited vision. They have also described the main character's features and the challenging social situations included in the game, which enhances the game's relevance and potential impact.

Thank you very much for your positive feedback.

Results and Discussion:

The discussion has been expanded to align with the principles of inclusive education and social learning theory. The authors have provided practical implications and recommendations for developers of digital educational tools, enriching the interpretation of the results and offering a useful guide for future developments in the field.

Thank you very much for your positive feedback.

Conclusions and Future Research:

It has been clarified how Vi-Connect addresses the deficiencies identified in the theoretical framework, providing a clearer connection between the results and the study's objectives. The authors have explicitly acknowledged the limitations related to the study design, such as sample size and geographical scope, and have proposed longitudinal studies to evaluate Vi-Connect's effectiveness over time.

Thank you very much for your positive feedback.

Additional Recommendations

Despite the significant improvements made, I suggest the following minor modifications for further clarity and coherence.

Terminology Clarification:

Ensure that the terminology used to describe individuals with visual impairments is consistent throughout the manuscript. For example, use terms like "visual impairment" or "limited vision" uniformly.

Thank you very much for your comment. We would like to clarify that the terms “vision impairment” and “visual impairment” are used interchangeably in the UK to describe to describe vision loss that cannot be corrected by glasses or contact lenses. However, the term “vision impairment” is mostly used by the VI community, therefore we decided to proceed with this terminology. Furthermore, according to the NHS classification system, people with vision impairment in the UK can be classified as either severely sight impaired (blind) or sight impaired (partially sighted or with low vision); please see here for more information The criteria for certification | RNIB. The students who participated in our two online focus groups (before and after the development of the prototype) were registered as sight impaired (partially sighted) at the time of the study, therefore we need to make this distinction in our manuscript as it may affect our findings and the development of the intervention. We have discussed this further in the limitation paragraph where we have previously added the following: “Third, as previously discussed, the focus of the current game was adolescents with sight impairment meaning that the players had some residual vision. Thus, more adaptations are required for student populations with more severe types of sight impairment.”

Game Implementation Details:

Include more details on how the game will be implemented in real educational settings. This could involve information about the necessary training for teachers and technical support to ensure effective integration.

Thank you for this very helpful comment. Additional information has been added to the last paragraph of the manuscript on page 36, which explains that the proposed intervention aims to promote inclusivity by being both user and family friendly: “In order for teachers and other school staff (e.g., teaching assistants and special educators) to promote Vi-Connect in the classroom, they do not need specialised training but access to a Windows device and an Internet connection. Students who wish to engage with Vi-Connect also do not need to receive special training or have previous experience playing video games, as there are detailed instructions and support (e.g. Games Master figure) to guide and help them to complete the intervention and also boost their confidence. The proposed intervention aims to promote inclusivity by being both user and family friendly.”

Citations and References:

Review and ensure that all citations and references are accurate and complete. Some minor typographical errors and inconsistencies in references should be corrected to maintain the academic accuracy of the manuscript.

Thank you for the helpful comment and suggestion. We have now checked and updated our reference list.

Impact Evaluation:

Include a brief discussion on how the long-term impact of the game will be measured beyond the initial evaluations. This will help contextualize the game's effectiveness and sustainability in different educational settings.

Thank you for this very helpful suggestion. Additional information has been added on page 35 to further explain how the long-impact of Vi-Connect can be tested in more diverse populations: “Our aim is to evaluate the effectiveness of Vi-Connect and trial it with different student groups with and without additional sensory impairments (e.g., hearing impairment and multi-sensory impairment or deafblindness) in different geographical locations and schools. To trial Vi-Connect with different populations and assess its long-term impact, our research team is considering a range of quantitative and qualitative research methodologies that can be tailored to the learning needs of each student population, such as interviews and focus groups, large surveys and observational research. We are interested in conducting longitudinal studies with diverse group of students examining the effect of such digital interventions on their social competence, school engagement and academic performance. The proposed research methods will help us understand whether students attending more well-resourced schools are more likely to have positive mental health and social outcomes compared to young people from deprived areas. Applying a range of different research methods may further promote mixed methods approaches to the development of educational interventions for students with SEND as well as open the research field to designing more effective interventions for students with sensory impairments who have been under-represented in previous research.”.

Conclusion

Overall, the authors have adequately addressed all the reviewers' comments and suggestions. The changes made have significantly improved the clarity, coherence, and methodological rigor of the manuscript. With the additional minor modifications recommended, I suggest that the revised manuscript be considered for publication in PLOS ONE.

Thank you very much for all your very positive feedback and suggestions.

Reviewer #2: Thank you for this resubmission. I have now had the opportunity to read through the revised manuscript. I am happy that the suggestions that I have made have been addressed.

Thank you very much for all your very positive feedback and suggestions.

---

## [Decision Letter · Decision Letter 2]

25 Jun 2024

The Development of Vi-Connect: An Educational Game for the Social Inclusion at School of Students With Vision Impairment

PONE-D-24-06103R2

Dear Dr. Manitsa,

We’re pleased to inform you that your manuscript has been judged scientifically suitable for publication and will be formally accepted for publication once it meets all outstanding technical requirements.

Kind regards,

Tien-Chi Huang

Academic Editor

PLOS ONE

Additional Editor Comments (optional):

After considering the reviewer’s comments, I agree to accept this study for publication in PLOS ONE, with the understanding that the authors will address the final suggestion provided by the reviewer.

The reviewer noted that while the manuscript has seen significant improvements, a minor edit of the English language by a native speaker is recommended to enhance clarity and readability. Therefore, I request that the authors undertake this final adjustment to ensure the manuscript meets the high standards of clarity and readability expected by the journal.

Once this revision is completed, I believe the manuscript will be well-suited for publication.

Reviewers' comments:

Reviewer's Responses to Questions

**Comments to the Author**

1. If the authors have adequately addressed your comments raised in a previous round of review and you feel that this manuscript is now acceptable for publication, you may indicate that here to bypass the “Comments to the Author” section, enter your conflict of interest statement in the “Confidential to Editor” section, and submit your "Accept" recommendation.

Reviewer #1: All comments have been addressed

2. Is the manuscript technically sound, and do the data support the conclusions?

Reviewer #1: Yes

3. Has the statistical analysis been performed appropriately and rigorously? 

Reviewer #1: Yes

4. Have the authors made all data underlying the findings in their manuscript fully available?

Reviewer #1: Yes

5. Is the manuscript presented in an intelligible fashion and written in standard English?

Reviewer #1: Yes

6. Review Comments to the Author

Reviewer #1: Dear authors,

Dear Authors,

Thank you for addressing and improving your manuscript based on the reviewers' comments. I believe that your manuscript has improved considerably in the following respects.

Confirmation of Addressed Changes:

Theoretical Framework and Literature Review: The authors expanded the review to include comparative studies of similar tools for populations with other disabilities and added a paragraph on the impacts of the COVID-19 pandemic on the education of students with visual impairments, providing a stronger context for the research.

Study Objectives: The objectives have been refined to include specific goals related to the socio-emotional skills that Vi-Connect aims to develop, enhancing clarity and precision.

Methodology: The participant selection process and inclusion/exclusion criteria have been detailed, improving transparency and methodological validity. Measures to ensure the game's accessibility and usability for participants with visual impairments have been described, ensuring equitable participation.

Results and Discussion: The discussion has been expanded to align with inclusive education principles, providing practical implications and recommendations for developers of digital educational tools.

Conclusions and Future Research: The conclusions have been clarified to show how Vi-Connect addresses theoretical deficiencies, with proposed longitudinal studies to evaluate the game's long-term effectiveness.

Minor Recommendations:

While the manuscript has been significantly improved, I recommend a minor edit of the English language by a native speaker to enhance clarity and readability. With this final adjustment, I suggest that the revised manuscript be considered for publication.

Sincerely,

David Pérez Jorge

7. PLOS authors have the option to publish the peer review history of their article (what does this mean?). If published, this will include your full peer review and any attached files.

Reviewer #1: **Yes: **David Pérez-Jorge

---

## [Editor Report · Acceptance letter]

25 Oct 2024

PONE-D-24-06103R2 

PLOS ONE

Dear Dr. Manitsa, 

I'm pleased to inform you that your manuscript has been deemed suitable for publication in PLOS ONE. Congratulations! Your manuscript is now being handed over to our production team.

Kind regards, 

on behalf of

Professor Tien-Chi Huang 

Academic Editor

PLOS ONE